# Experimental Study on the Drag Reduction Performance of Clear Fracturing Fluid Using Wormlike Surfactant Micelles and Magnetic Nanoparticles under a Magnetic Field

**DOI:** 10.3390/nano11040885

**Published:** 2021-03-31

**Authors:** Ming-Liang Luo, Xiao-Dong Si, Ming-Zhong Li, Xiao-Han Jia, Yu-Ling Yang, Yong-Ping Zhan

**Affiliations:** 1Key Laboratory of Unconventional Oil and Gas Development, China University of Petroleum (East China), Ministry of Education, Qingdao 266580, China; yfsailing_wxg@163.com (M.-L.L.); sixiaodong0021@163.com (X.-D.S.); 13793218982@163.com (X.-H.J.); yangyl202103@163.com (Y.-L.Y.); zhanyongping@upc.edu.cn (Y.-P.Z.); 2College of Petroleum Engineering, China University of Petroleum (East China), Qingdao 266580, China

**Keywords:** magnetic nanoparticles, wormlike micelles, fracturing fluids, magnetic field, drag reduction

## Abstract

This paper examines a new study on the synergistic effect of magnetic nanoparticles and wormlike micelles (WLMs) on drag reduction. Fe_3_O_4_ magnetic nanoparticles (FE-NPs) are utilized to improve the performance of viscoelastic surfactant (VES) solutions used as fracturing fluids. The chemical composition and micromorphology of the FE-NPs were analyzed with FT-IR and an electron microscope. The stability and interaction of the WLM-particle system were studied by zeta potential and cryo-TEM measurements. More importantly, the influences of the temperature, FE-NP concentration, magnetic field intensity, and direction on the drag reduction rate of WLMs were systematically investigated in a circuit pipe flow system with an electromagnetic unit. The experimental results show that a suitable content of magnetic nanoparticles can enhance the settlement stability and temperature resistance of WLMs. A magnetic field along the flow direction of the fracturing fluid can improve the drag reduction performance of the magnetic WLM system. However, under a magnetic field perpendicular to the direction of fluid flow, an additional flow resistance is generated by the vertical chaining behavior of FE-NPs, which is unfavorable for the drag reduction performance of magnetic VES fracturing fluids. This study may shed light on the mechanism of the synergistic drag reduction effects of magnetic nanoparticles and wormlike micelles.

## 1. Introduction

In the past few decades, conventional oil and gas production has been unable to meet the world’s growing demand for oil and gas resources, and unconventional oil and gas resources have become the focus of global attention [1,2,3,4]. At present, the key technologies of unconventional oil and gas exploitation are horizontal well drilling and volume (or network) fracturing technology [5,6,7]. Due to the tightness of unconventional reservoirs, a large amount of turbulent pipeline friction is generated in the process of volume fracturing, which will greatly increase the necessary hydraulic horsepower, damage the pumping equipment, and even cause failure of fracturing. Therefore, the drag reduction agent (DRA) in the fracturing fluid is one of the important materials to improve volume fracturing for the stimulation of unconventional reservoirs [8,9]. The addition of a DRA not only reduces the friction on fracturing lines and improves the complexity of fractures but also reduces the required hydraulic horsepower of equipment and minimizes wear caused by high-speed impact in the operation processes of equipment.

It is well known that turbulence inside the transported fluid brings about a large amount of frictional resistance [10]. Frictional resistance can be considerably reduced by adding only a minute amount of selected materials, known as DRAs, to the system [11,12]. These DRAs have the ability to reduce energy consuming and pumping power and increase piping system capacity. Therefore, DRAs are particularly gaining tremendous attention in different industrial areas, such as hydraulic fracturing [13,14], oil transportation pipelines [15], and district heating and cooling [16]. Polymers (such as polyacrylamides, polyethylene oxide, and guar gum) [17,18] and surfactants (such as cetyltrimethylammonium chloride and oleyldimethylamine oxide) [19,20] have been widely used as DRAs. In recent years, polyacrylamide and its derivatives have been widely used in hydraulic fracturing as DRAs [21]. However, their performance metrics, such as their temperature and salt resistance, mechanical shear resistance, and reservoir damage, cannot meet the requirements of hydraulic fracturing for unconventional oil and gas reservoirs [22]. The characteristics of surfactant wormlike micelles (WLMs), such as their small molecular weight, reversible resistance reduction, recoverable mechanical degradation, ability to break down gels without leaving a residue, and environmental friendliness, are widely considered to be beneficial [23]. Surfactants form a wormlike three-dimensional network structure under the action of counterionic salts and other substances, and have the property of viscoelasticity. According to an experimental study by Li [24], a small amount of surfactant added to the turbulent fluid can reach a drag reduction rate of up to 80%. Compared with Newtonian fluids, surfactant fluids contribute to viscoelastic shear stress, suppress turbulent vortex structures, reduce turbulent shear stress, and, thus, reduce friction resistance. Surfactant molecules form highly entangled wormlike micelles in the presence of brine. The micelle structure builds viscoelasticity in fracturing fluids and transports proppants in pipelines and fractures [25]. The entangled structure then breaks when exposed to hydrocarbons or diluted with formation water, and the gel-breaking solution does not harm formation [26]. However, wormlike micelle solutions have the disadvantages of weak stability and low viscosity at temperatures above 90 °C [27], leading to the failure of drag reduction in hydraulic fracturing.

With the continuous development of nanotechnology, some progress has been made in the application of SiO_2_, TiO_2_, Al_2_O_3_, and carbon nanotubes to improve the stability and viscoelasticity of WLMs [28,29,30]. Nettesheim et al. [31] added 1.0 vol% SiO_2_ nanoparticles to WLMs, with NaNO_3_/cetyltrimethylammonium bromide (CTAB) as the fracturing fluid, and found that nanoparticles can effectively increase the viscosity of WLMs. Huang and Crews [32] witnessed an improvement in viscosity after adding 0.12~0.18% nanoparticles to WLMs with 2% surfactant. Luo et al. [33] studied the effect of pyroelectric BaTiO_3_ nanoparticles on WLMs and found an effective enhancement of temperature resistance and viscoelasticity. Philippova and Molchanov [34] found that the pseudo-crosslinking effect between nanoparticles and micelles can increase the viscosity of WLMs by three orders of magnitude and improve their temperature resistance. Wu et al. [35] modified WLMs with 0.01% SiO_2_ nanoparticles, improving temperature resistance and sand-suspending capacity. However, conventional nanoparticles have no response to external stimuli, so the flow characteristics and distribution of nanoparticles in the pores or fissures of rock cannot be regulated by external stimuli. Moreover, these nanoparticles are easily adsorbed on the surface of rock and difficult to separate from oil and water after flowback from the formation, increasing the engineering application risks of nanofluids.

Recently, the application of nanomagnetic fluids in the petroleum industry has attracted much attention. Nanomagnetic fluids are stable colloidal liquids composed of magnetic solid particles with nanometer diameters and magnetic response abilities, and they have nonmagnetic liquids as carriers and dispersants [36]. First, magnetic fluids show the characteristics of general fluids, and their motion follows the law of fluid dynamics. Second, magnetic fluids are a magnetic substance whose behavior is controlled by a magnetic field. Therefore, magnetic nanoparticles are easily separated from the dispersion liquid by an external magnetic field and, thus, can be reused [37,38]. At present, research on nanomagnetic fluids in oil and gas production mainly focuses on reservoir fracture identification [39], oil-water front monitoring [40], and oil recovery improvement [41,42,43,44], while some scholars have carried out preliminary investigations of drag reduction by nanomagnetic fluids. Sun et al. [45] designed a magnetic viscous fluid drag reduction device with a closed loop by placing a permanent magnet outside the tube wall. The magnetic fluid forms a viscous fluid layer on the inner wall of the tube, leading to a drag reduction effect. Rosa et al. [46] studied the drag reduction characteristics of a nanomagnetic fluid under an asymmetric magnetic field by a numerical simulation method and found that the magnetic field gradient has a great influence on the viscosity and friction coefficient of the nanomagnetic fluid. Nadia et al. [47] prepared magnetic hydrophobic nanoparticles of m-TiO_2_@Cys and studied their flow characteristics in a pipeline, in which the drag reduction rate increased by 12%. However, only a few published studies applied magnetic nanoparticles for drag reduction, and those that have done so failed to perform experimental verification and mechanistic analysis. Additionally, previous studies were conducted at room temperature without a magnetic field, which does not accurately represent the requirements of fracturing under high shear and high-temperature environments.

Accordingly, this work aims to gain further insights into the role and mechanism of wormlike micelles with magnetic nanoparticles in drag reduction. First, the magnetic nanoparticles were chemically modified with sodium oleate and characterized by an electron microscope and FT-IR. Then, magnetic WLMs as fracturing fluids were prepared by the interactions between magnetic nanoparticles and micelles. The stability and interaction of the WLM-particle system were studied by zeta potential and cryo-TEM measurements. More importantly, the influences of the temperature, FE-NP concentration, and magnetic field intensity and direction on the drag reduction rate of WLMs were systematically investigated in a circuit pipe flow system with an electromagnetic unit.

## 2. Materials and Methods

### 2.1. Materials

Ferric oxide nanoparticles (FE-NPs) were obtained (Sigma-Aldrich, Sigma-Aldrich (Shanghai) Trading Co., Ltd., Shanghai, China, d50 ≅ 20 nm). Sodium oleate, potassium chloride, and ethanol with purities of ≥99 wt%, ≥99 wt% and ≥99.5 wt%, respectively, were manufactured by Macklin Biochemical Technology Co., Ltd., Shanghai, China. Bactericide JUN-1 and defoamer DT-882D were provided by PetroChina Changqing Downhole Technical Operation Company, Xi’an, China.

### 2.2. Surface Modification of FE-NPs

To prepare magnetic WLMs with good stability as fracturing fluids, FE-NP surface modification was required. First, 0.3 g of FE-NPs was placed in a three-mouth flask, and 180 mL of anhydrous ethanol was added under ultrasonic dispersion for 30 min. The mixture was placed in a water bath at 60 °C and stirred by mechanical agitation for 1 h. Then, 10 mL sodium oleate was added dropwise to the reaction solution over 30 min. The reaction was stirred mechanically at 60 °C for 2 h. After cooling to room temperature, the products were separated by an external magnetic field, washed repeatedly with deionized water until a neutral pH was reached, and dried in vacuum at 60° C for 12 h. Then, sodium oleate-modified FE-NPs were obtained.

### 2.3. Preparation of Magnetic WLM Fracturing Fluid

The sodium oleate-modified FE-NPs were added gradually to a sodium oleate solution with a pH value of 8.5 at 50 °C and stirred for 60 min at 300 r/min, until they were well combined. Then, the counterion salt KCl with a concentration of 2.0 wt%, bactericide JUN-1 with 0.2 wt%, and defoamer DT-882D with 0.5 wt% were added to the mixed solution, which was stirred for an additional 30 min and dispersed for 15 min by ultrasonic vibration. The magnetic WLMs as a fracturing fluid were obtained after standing for 24 h at room temperature.

### 2.4. Characterizations of the FE-NPs and the Magnetic WLM System

The main functional groups of FE-NPs were verified by FT-IR with a Nicolet 6700 FT-IR instrument (Thermo Fisher Scientific Inc., Waltham, Mass., USA). The FT-IR sample was prepared by the potassium bromide (KBr) pellet method. The surface morphology of the FE-NPs was observed by TEM and FE-SEM. TEM images of the FE-NPs were obtained with a JEOL microscope (JEM-2100, JEOL Ltd., Toyoshima, Tokyo, Japan). The SEM images of the sample surface were also obtained using a MIRA 3 series scanning electron microscope equipped with a Schottky field emission electron gun.

Without an external magnetic field, the viscosity of the VES fracturing fluid without FE-NPs was 26.4 mPa·s, at a shear rate of 170 s^−1^ at 25 °C (tested with an Anton Paar rheometer, Physica MCR 301, Anton Paar GmbH, Graz, Austria). The dispersion stability of magnetic WLMs was studied by zeta potential measurements with a Zetasizer (Nano ZS, Malvern instruments Ltd, Malvern, UK).

The microstructure of the magnetic WLMs was observed with cryo-TEM. All the samples for cryo-TEM observation were kept at 25 °C. Then, we took out 3 μL of the sample to place on a TEM copper grid, and obtained a thin liquid film with two pieces of filter paper. Next, we rapidly put the grid into a cryogen reservoir of liquid ethane at −175 °C and stored it with liquid nitrogen (−180 °C). Then, the frozen specimen was transferred into a cryo-holder and was observed with cryo-TEM (Krios G4, Thermo Fisher Scientific Inc., Waltham, MA, USA). Finally, the images were recorded digitally with a charge-coupled device camera and conducted with Digital Micrograph software. 

In addition, the orientation of magnetic nanoparticles in micellar solution under a magnetic field was observed by optical microscopy (M330-3M180, AOSVI, Shenzhen Aosvi Optical Instrument Co., Ltd., Shenzhen, China).

### 2.5. Test Facility

The drag reduction performances of magnetic WLMs as fracturing fluids were measured with a closed loop, as shown schematically in Figure 1. The system consisted of a storage tank installed with a 5 kW heater, a stainless steel pipeline with an inner diameter of 5 mm, a length of 3 m, an inner wall roughness of 0.01524 mm, a screw pump with a maximum flow rate of 0.4 m^3^/h (measuring accuracy of 0.001 m^3^/h), an electromagnetic flowmeter (Jiangsu Leitai Automation Instrument Engineering Co., Ltd., Huaian, China), a differential pressure transmitter, and a unit with a generating magnetic field. The unit with a generating magnetic field was composed of a horizontal magnetic field generator and a vertical magnetic field generator. The horizontal magnetic field generator was a solenoid (a length of 1 m, range of 0~0.6 T) and could provide uniform magnetic field along the pipeline. The magnetic field intensity of the solenoid was controlled with a DC power supply (range of 0~20 A, RD-S30020, Suzhou Varied Electric Co., Ltd., Suzhou, China) and could be monitored by a Gauss meter (range of 0~3 T, CH-1500, Beijing CH-Magnetoelectricity Technology Co., Ltd., Beijing, China). The vertical magnetic field generator (range of 0~0.6 T, CH-TUD-300-100, Beijing CH-Magnetoelectricity Technology Co., Ltd., Beijing, China) was controlled by a DC power supply and could provide a uniform magnetic field perpendicular to the pipeline.

Prior to the start of the measurements, the pipeline was subjected to freshwater cleaning. The electrical heater controlled the temperature of the storage tank. The magnetic WLM fracturing fluid was placed in the storage tank. Next, the power was switched on from the cabinet control panel, and the screw pump was activated to circulate the magnetic WLMs in the pipeline. Every time the flow rate was adjusted, the fluid circulated in the pipeline for 3–5 min. After the pressure data shown on the control cabinet panel were stabilized, the data were recorded. Each group of measurements was repeated at least three times at an interval of 2 min to eliminate operational error. The average value of the data for the three trials was taken as the pressure dropped at each flow rate, and then the flow rate was increased and the test was continued until completion. In the experiment, the magnetic field intensity was adjusted by changing the current. The percentage of the drag reduction rate (%*DR*) is calculated as follows:*%DR* = ((∆*P_w_* − ∆*P*_0_)/∆*P_w_*) × 100(1)
where Δ*P_w_* and Δ*P*_0_ (both in Pa) refer to the pressure drops produced by the fresh water and magnetic WLM as the fracturing fluid, respectively.

In addition, the contact angle measurement of water on stainless steel treated by magnetic nanoparticles was carried out by an automatic contact angle meter model JCY Series (Shanghai Fangrui Instrument Co., Ltd., Shanghai, China) to indicate that the modified magnetic nanoparticles could change the wettability of the stainless steel surface.

## 3. Results and Discussions

### 3.1. Chemical Groups and Micro Morphology of the FE-NPs

Figure 2 shows the FT-IR spectra of the FE-NPs and surface-modified FE-NPs. The peaks at approximately 571 cm^−1^ and 1632 cm^−1^ represent the vibration peak of the Fe-O bond and the multiple vibration peaks of the Fe-O bonds in all the IR spectra, respectively, confirming the existence of the FE-NPs. The hydroxyl -OH stretching peaks appear at approximately 3407 cm^−1^ in all the IR spectra, indicating that the polar surface of the FE-NPs adsorbed water molecules. Compared with those of the untreated FE-NPs (Figure 2a), there are some additional vibration peaks in the IR spectrum of the treated FE-NPs (Figure 2b), i.e., a carboxyl -COO- stretching vibration peak at 1562 cm^−1^ and stretching vibration peaks of the C=C bond and -CH_2_- and -CH_3_ groups at 1413 cm^−1^, 2854 cm^−1^, and 2918 cm^−1^, respectively; these indicate that sodium oleate was attached to the surface of the FE-NPs by covalent bonding between the carboxylate moiety and the Fe atom [48].

The micro-morphologies of the untreated FE-NPs and FE-NPs treated with sodium oleate were examined by SEM and TEM (as seen in Figure 3). It is known that pure iron has two valence states in Fe_3_O_4_ with an inverse spinel structure, and the shape of crystals is mostly determined by the relative growth rates along different directions. Moreover, the alkaline solution and surfactant promote the growth of various morphologies of metal-oxide crystals [49]. As seen from the SEM images, the untreated FE-NPs appear to exhibit an octahedral structure (seen in Figure 3a), while the treated FE-NPs show an almost spherical-like morphology due to the coated surfactant layer (seen in Figure 3b). Additionally, the TEM image of the treated FE-NPs (Figure 3c) shows that the average particle size is approximately 20 nm, and there is no obvious agglomeration between the particles, which is consistent with the results shown in Figure 3b.

The zeta potential, as the potential of a shear plane, is an important index used to characterize the stability of colloidal dispersion systems [50]. The zeta potentials of WLMs with different concentrations of FE-NPs were measured, and the results are shown in Figure 4. The zeta potentials of WLMs are affected by the concentration of nanoparticles [51,52]. Both the untreated and the treated FE-NPs exhibited a similar trend in zeta potential as the concentration of FE-NPs in the WLMs increased. When the FE-NP concentration was very low or high, the absolute value of the zeta potential of the WLMs was relatively small, and the absolute values of the zeta potential of the WLMs with the treated FE-NPs were significantly higher than those of the WLMs with the untreated FE-NPs. When the absolute value of the zeta potential is greater than 45 mV, the dispersed suspension system is relatively stable [52]. Here, the maximum zeta potential of the WLMs with treated FE-NPs was 49.9 mV, indicating that the WLMs with the treated FE-NPs were very stable.

In addition to the zeta potential measurement, we also used cryo-TEM to study the microstructure and particle distribution in the WLMs with and without FE-NPs. Note that cryo-TEM is a technique where the structure in a fluid sample is preserved by rapid freezing in a controlled environment. As shown in Figure 5a–d, WLMs (marked by the white arrow) were observed in all the samples with and without FE-NPs (marked by the red circle). It is also clearly seen that, with the addition of FE-NPs, the WLMs were entangled with each other and appeared to intersect with the FE-NPs, as shown in Figure 5b,c. Simultaneously, the aforementioned phenomena further verify the hypothesis given by reference [34] that the self-assembly of nanoparticles and micelles forms a WLM-nanoparticle three-dimensional network structure system and have the property of viscoelasticity. The micellar network absorbs and stores the turbulent kinetic energy of micellar solution in high-shear-rate region. When the high-shear-rate region diffuses or moves to the low-shear-rate region, the energy stored in the micellar network structure is released into the solutions in the form of large-scale vortexes [53]. The processes of storing and releasing processes of energy reduces dissipation of turbulent kinetic energy and result in energy-saving benefits, then, the drag reduction occurs [54]. However, when the nanoparticle concentration was too high, the FE-NPs agglomerated and inhibited the formation of the WLM structure, as displayed in Figure 5d. The observed microstructure of the samples with different FE-NP concentrations in the cryo-TEM images could give us a better understanding of the special rheological properties of WLM-nanoparticle systems.

### 3.2. Drag Reduction Performance of WLMs as Fracturing Fluids

#### 3.2.1. Effects of the Temperature and FE-NP Concentration

As shown in Figure 6, in the experimental temperature range (25~40 °C), a WLM network structure of viscoelastic surfactant fracturing fluid (VES-FF) solution without FE-NPs was formed and had a certain temperature resistance and shear stability; as such, the drag reduction rate (DR) remained high with increasing temperature. When the temperature is high (>40 °C), the drag reduction rate decreases rapidly with increasing temperature, because the structure of WLMs formed by the self-assembly of surfactant molecules is gradually destroyed and transformed into rod-shaped micelles and spherical micelles, due to violent molecular motion. When the temperature exceeds 80 °C, the drag reduction rate is less than 10%; that is, the temperature increases from 25 °C to 90 °C, and the loss of the drag reduction rate is as high as 89.5%. Viscoelasticity decreases and friction resistance increases under turbulence, which shows that the drag reduction rate decreases rapidly.

When the concentration of FE-NPs is low (0.1~0.3 wt%), the drag reduction rate of the particle-micelle solution is higher than that of the VES solution without FE-NPs at the same temperature, which indicates that there is a synergistic drag reduction effect between magnetic nanoparticles and surfactant WLMs. FE-NPs and WLMs form a particle–micelle composite network structure, which increases the complexity of the WLM structure and improves the temperature and shear resistance of the system. Therefore, the drag reduction of magnetic VES-FF decreases slowly with increasing temperature, and the drag reduction rate of VES-FF with 0.3 wt% FE-NPs is still as high as 35.2% at high temperature (90 °C). However, at high temperature (>80 °C), the drag reduction ability of VES-FF with 0.3 wt% FE-NPs is significantly better than that of VES-FF with 0.1 wt% FE-NPs. The network structure of WLMs is almost destroyed at high temperature, and there is a sharp decline in the drag reduction ability of magnetic VES-FF. However, the magnetic nanoparticles modified by sodium oleate, which originally play the role of cross-linking micelles, are stripped from the VES-FF system and are attached to the inner wall of the stainless steel pipeline, which can change the wettability of the wall. As shown in Figure 7a, with increasing FE-NP concentration, the wetting angle of the stainless steel surface increases gradually, even exceeding 136°, forming a layer of superhydrophobic nanoparticle film which can weaken the turbulent vortex and reduce the energy convection loss and drag (as shown in Figure 7b); these results are consistent with the findings of previous studies [47].

When the concentration of FE-NPs is high (0.5 wt%) and the temperature is low (<55 °C), the drag reduction effect of the WLM solution with FE-NPs is even worse than that of the WLM solution without FE-NPs due to the agglomeration of some nanoparticles. However, when the temperature is high (>60 °C), the drag reduction rate of the system begins to slow down and even increases, which is due to the adhesion in the tube forming a superhydrophobic nanoparticle film layer on the wall surface, which can reduce drag. Therefore, magnetic nanoparticles and wormlike micelles have synergistic drag reduction effects. There is a critical concentration of magnetic nanoparticles (0.3 wt%) that can maximize the temperature resistance, shear resistance, and drag reduction performance of magnetic VES-FF.

#### 3.2.2. Effect of the Horizontal Magnetic Field Intensity

At 25 °C, a horizontal magnetic field (along the flow direction of the fracturing fluid) is applied to the friction test pipeline. Fe_3_O_4_ nanoparticles possess superparamagnetic characteristics, leading them to easily respond to magnetic field. Therefore, in order to study the influence of magnetic field intensity on the drag reduction performance of the system, the magnetic field intensity is slowly increased with the variation increment of 0.05 T, at the initial stage. As shown in Figure 8a, when the magnetic field intensity is weak (B < 0.1 T), the drag reduction ability of magnetic VES-FF first increases and then decreases. When C_FE-NPs_ = 0.3 wt% and B = 0.055 T, the drag reduction rate of the system is as high as 80.8%, indicating that a low magnetic field intensity can improve the drag reduction performance of magnetic VES-FF in a turbulent state. As seen from Figure 8c,d, on the one hand, the magnetic nanoparticles, coupled with WLMs, can improve the stability of the network structure and enhance the drag reduction stability of the system; on the other hand, the horizontal magnetic field can add a horizontal drag force along the flow direction to the magnetic particle–micelle system, which can not only improve the viscoelasticity of the system but also enhance the fluidity of the system to reduce turbulent friction and improve drag reduction performance. When B = 0.055 T, the drag reduction rate reaches a maximum and gradually drops off, exceeding the value of 0.055 T. Therefore, magnetic field intensity is rapidly increased with the variation increment of 0.1 T at the following stage. As shown in Figure 8b, when the magnetic field intensity is strong (B > 0.1 T), the drag reduction ability of magnetic VES-FF is obviously worse. When C_FE-NPs_ = 0.3 wt% and B = 0.5 T, the drag reduction rate of the system is as low as 40.1%. Compared with B = 0.055 T, the loss of the drag reduction rate of the system is more than 50.4%, which indicates that, under a turbulent state (high flow velocity, strong shear), the strong magnetic field will reduce the drag reduction effect of the system. Figure 8e reveals that, under the action of a strong magnetic field, the magnetic nanoparticles will chain and coarsen, and even agglomerate and settle, which will destroy the particle-WLM structure of magnetic VES-FF and increase the flow resistance. Therefore, the existence of a critical horizontal magnetic field intensity (B = 0.055 T) can enable the magnetic VES-FF system to maintain the best drag reduction effect and the best drag reduction stability.

#### 3.2.3. Effect of the Vertical Magnetic Field Intensity

As shown in Figure 9a, a magnetic field perpendicular to the fluid flow direction cannot improve the drag reduction effect of the magnetic VES-FF system. During the process of applying a magnetic field, the influence of the vertical magnetic field intensity on the drag reduction rate of magnetic VES-FF can be divided into three slope regions. In region I (B = 0~0.02 T), when the magnetic field is very weak, the drag reduction rate remained unchanged, or the system did not noticeably respond to the magnetic field in this range. In region II (B = 0.02~0.1 T), with increasing magnetic field intensity, there was a sharp drop in the drag reduction rate. For C_FE-NPs_ = 0.3 wt%, the drag reduction rate of magnetic VES-FF under B = 0.1 T is reduced by 61.6% compared with that under B = 0.02 T. Therefore, the vertical magnetic field obviously had a counterproductive effect on drag reduction. These results can be explained by the action of a vertical magnetic field: magnetic nanoparticles were adsorbed on the inner surface of the tube wall and gradually chained (as shown in Figure 9b), forming a series of “chain doors”. On the one hand, these chain doors destroy the particle–micelle network structure and result in a large loss of the effective surfactant concentration for drag reduction. On the other hand, these chain doors increase the frictional resistance at the pipe wall and reduce fluid flow space. The fluid must push open or destroy these chain doors to flow freely, so the flow resistance increases, which manifests as a rapid decrease in drag reduction rate. In region III (B > 0.1 T), with a further increase in magnetic field intensity, the drag reduction rate decreases slightly and tends to be stable, reflecting that the chain resistance-increasing behavior of magnetic particles basically reaches its peak before B = 0.1 T. As a result, the magnetic field perpendicular to the direction of fluid flow is unfavorable for the drag reduction ability of magnetic VES-FF.

## 4. Conclusions

This paper systematically reported an investigation of the rules and mechanism of magnetic VES fracturing fluid on drag reduction. FE-NPs treated with sodium oleate were octahedral, with a uniform diameter of 20 nm, which could promote micelle growth and entanglement through the interactions between micelles and nanoparticles. The synergistic drag reduction mechanism of magnetic nanoparticles and surfactant wormlike micelles mainly includes three aspects. First, the WLM-nanoparticle three-dimensional network reduces dissipation of turbulent kinetic energy and brings about the drag reduction. Second, some of the treated FE-NPs attached to the inner wall of the pipeline form a layer of superhydrophobic nanoparticle film, leading to drag reduction. Finally, the magnetic field along the flow direction provides an additional drag force to enhance the fluidity of the system and reduce turbulent frictional resistance. There was an optimal FE-NP concentration that could effectively improve the stability, temperature resistance, and drag reduction performance of the magnetic WLM system. The results showed that a weak magnetic field (<0.055 T) along the flow direction of the fracturing fluid can improve the drag reduction performance of the magnetic WLM system, and the drag reduction rate is as high as 80.8% at 25 °C. However, a magnetic field that is too strong will obviously reduce the drag reduction effect of the system. Moreover, under a magnetic field perpendicular to the direction of fluid flow, additional flow resistance is produced by the vertical chaining behavior of FE-NPs, which is unfavorable for the drag reduction ability of the magnetic VES fracturing fluid. Finally, the magnetic VES fracturing fluid is a smart fluid that can respond to an external magnetic field and be controlled by equipment from the surface. Therefore, this study offers a cleaner and more efficient fracturing fluid for the future hydraulic fracturing of unconventional reservoirs.

## Figures and Tables

**Figure 1 nanomaterials-11-00885-f001:**
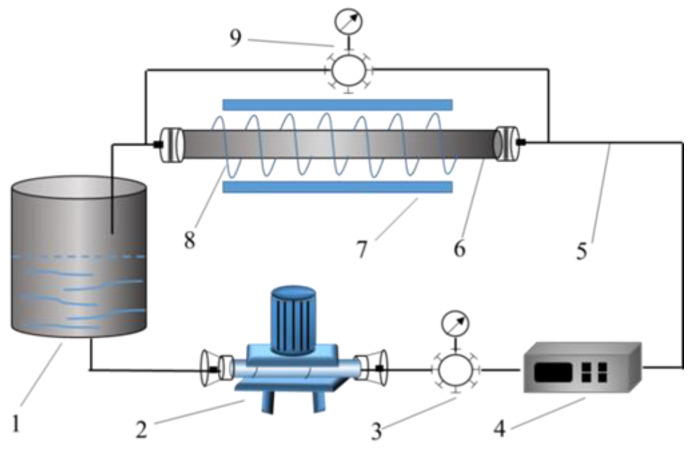
Schematic of the experimental system: (1) storage tank; (2) screw pump; (3) electromagnetic flowmeter; (4) control unit; (5) connection section of pipeline; (6) measurement section of pipeline; (7) electromagnet; (8) coil; (9) differential pressure transmitter.

**Figure 2 nanomaterials-11-00885-f002:**
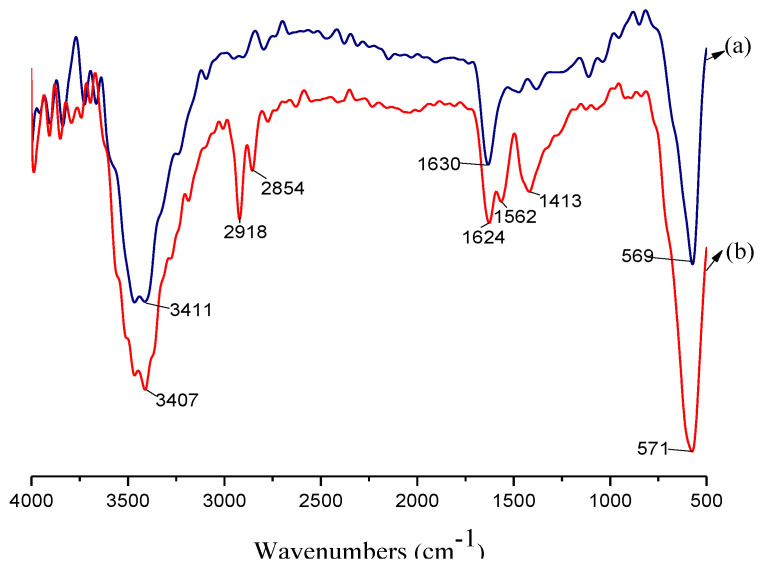
FT-IR spectra of the untreated (**a**) and treated (**b**) ferric oxide nanoparticles (FE-NPs) with sodium oleate.

**Figure 3 nanomaterials-11-00885-f003:**
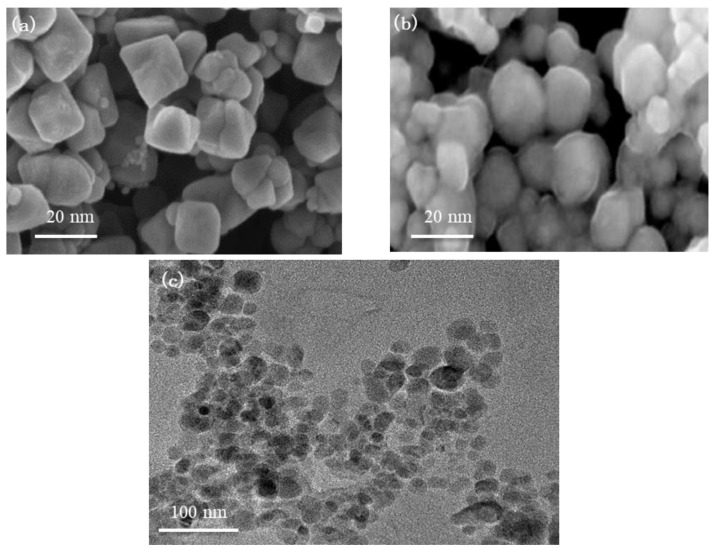
Micrographs of FE-NPs: (**a**) SEM image of untreated FE-NPs; (**b**) SEM image of FE-NPs treated with sodium oleate; and (**c**) TEM image of treated FE-NPs.

**Figure 4 nanomaterials-11-00885-f004:**
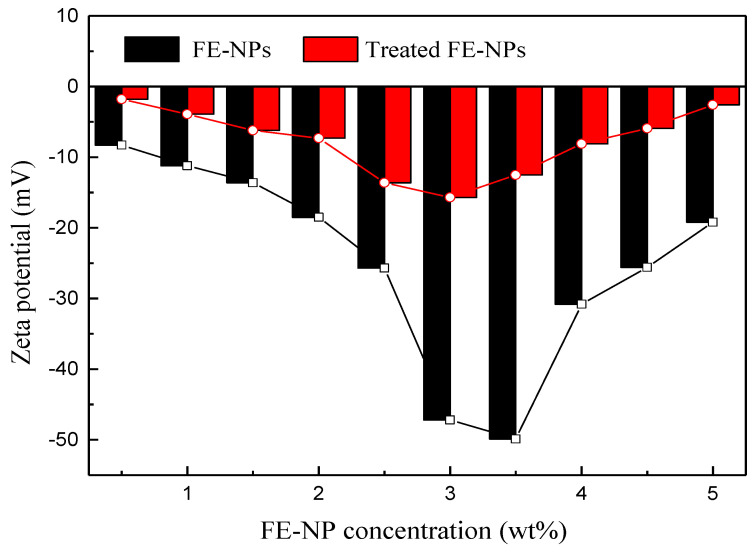
Zeta potentials of the wormlike micelles (WLMs) with different FE-NP concentrations.

**Figure 5 nanomaterials-11-00885-f005:**
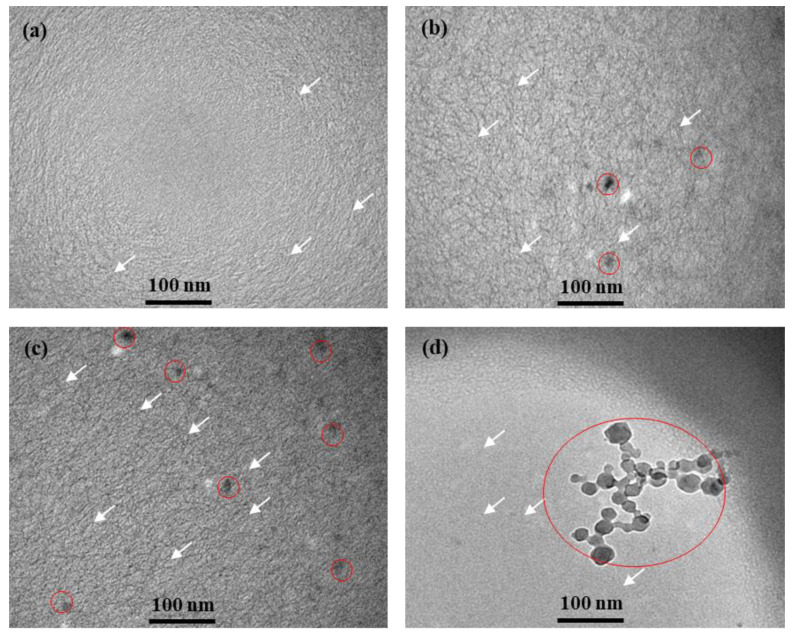
Cryo-TEM micrographs of the WLMs with different FE-NP concentrations: (**a**) 0 wt%; (**b**) 0.1 wt%; (**c**) 0.3 wt%; and (**d**) 0.5 wt%.

**Figure 6 nanomaterials-11-00885-f006:**
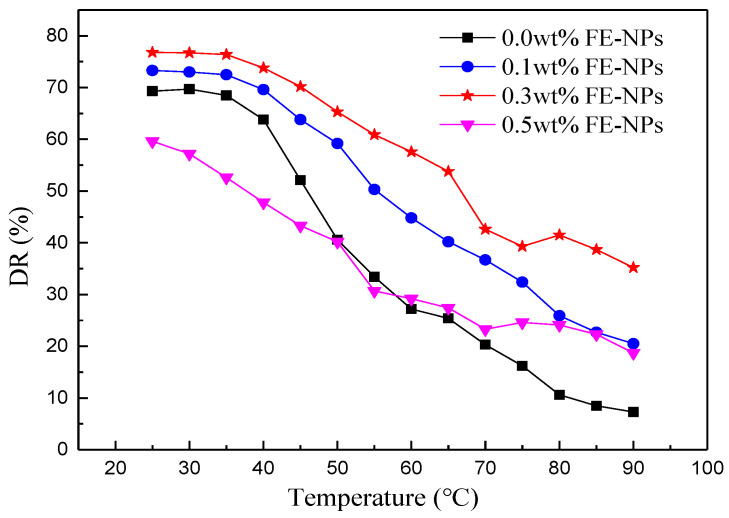
Effects of the temperature and FE-NP concentration on the drag reduction ability of VES-FF.

**Figure 7 nanomaterials-11-00885-f007:**
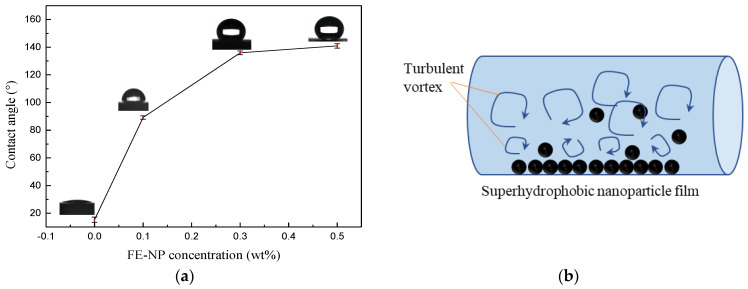
Drag reduction effect of FE-NPs: (**a**) water contact angle on the surface of stainless steel vs. the FE-NP concentration and (**b**) schematic diagram of drag reduction by superhydrophobic nanoparticles.

**Figure 8 nanomaterials-11-00885-f008:**
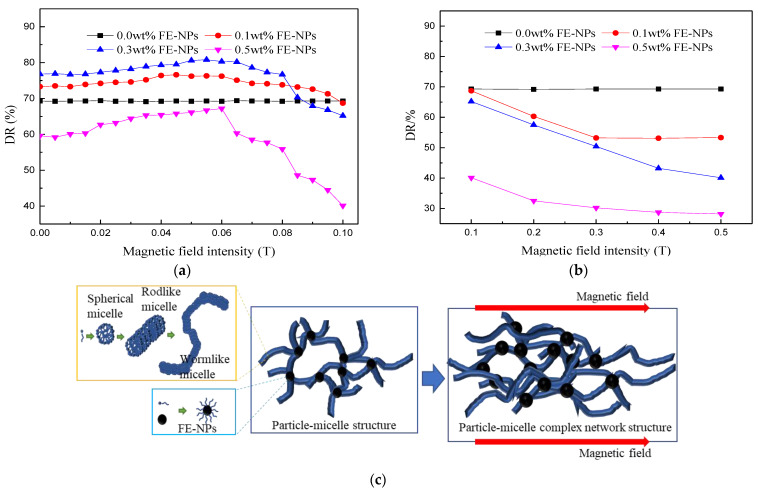
Drag reduction effect of a horizontal magnetic field on VES-FF at 25 °C: (**a**) Drag reduction rate (DR) vs. horizontal magnetic field intensity (B ≤ 0.1 T); (**b**) DR vs. horizontal magnetic field intensity (B > 0.1 T); (**c**) effect of the horizontal magnetic field on the particle–micelle network structure; (**d**) microphotograph with 500 times magnification of magnetic VES-FF with 0.3 wt% FE-NPs under a weak magnetic field (B = 0.055 T); and (**e**) microphotograph with 500 times magnification of magnetic VES-FF with 0.3 wt% FE-NPs under a weak magnetic field (B = 0.5 T).

**Figure 9 nanomaterials-11-00885-f009:**
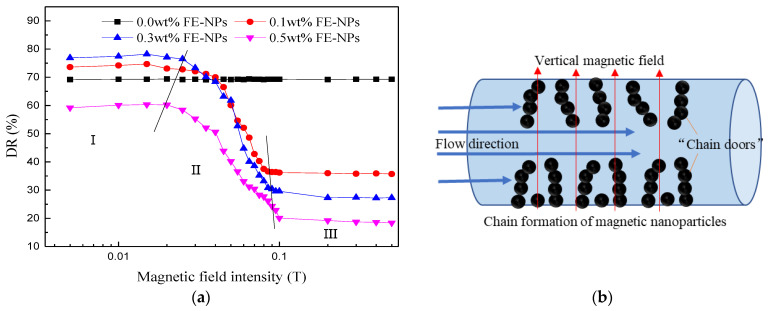
Effect of a vertical magnetic field on the drag reduction ability of magnetic VES-FF: (**a**) DR as a function of the vertical magnetic field intensity at 25 °C and (**b**) schematic diagram of the chain resistance increasing behavior of magnetic nanoparticles under a vertical magnetic field.

## Data Availability

Data are contained within the article.

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
