# Peer review of "Experimental Study on the Drag Reduction Performance of Clear Fracturing Fluid Using Wormlike Surfactant Micelles and Magnetic Nanoparticles under a Magnetic Field"

_nanomaterials, 2021, doi:10.3390/nano11040885_

Round 1
Reviewer 1 Report
The author presented a very interesting article entitled "Experimental study on the drag reduction performance of clear fracturing fluid using wormlike surfactant micelles and magnetic nanoparticles under a magnetic field”. In general, it is written and presented in an appropriate way. Hence, I am recommending for publication in “Nanomaterials” after minor revisions. Few points:
1. The mechanism of the synergistic drag reduction effect of magnetic nanoparticles is not clear in the paper. I would like to request the authors to elaborate in more detail.
2. I will encourage authors to mention more references on drag reduction agents (DRAs) reported recently in the literature.
3. How the magnetic field intensity has been chosen for this study as stated in figure 8?
4. Authors should correct some errors in the typos and references.
Author Response
The authors thank the reviewer for the excellent suggestions and valuable comments.
Please see the attachment.

Reviewer 2 Report
The paper "Experimental study on the drag reduction performance of clear fracturing fluid using wormlike surfactant micelles and magnetic nanoparticles under a magnetic field" by Ming-Liang Luo et al. is very interesting and clearly presented and I would like to recommend for publication in this Journal.
The Authors presented a well detailed introduction, materials and results carefully addressed.
More details on cryo-TEM technique and on the experimental setup to perform magnetic measurements would be very welcome to the readers.
Author Response
The authors thank the reviewer for the recognition and the excellent suggestions.
Please see the attachment.
